# Comparison of dose-adjusted EPOCH-R and R-CHOP in diffuse large B-cell lymphoma with high Ki67 expression: Results from a prospective observational study

**Wenxin Jiang, Peng Liu** *

Department of Medical Oncology, National Cancer Center/National Clinical Research Center for Cancer/ Cancer Hospital, Chinese Academy of Medical Sciences & Peking Union Medical College, Beijing, China

* liupeng@cicams.ac.cn

## Abstract

### Objectives

To compare the safety and efficacy of first-line dose-adjusted EPOCH-R (DA-EPOCH-R) versus R-CHOP treatment in diffuse large B-cell lymphoma (DLBCL) patients with high Ki67 expression.

### Methods

A retrospective analysis of DLBCL patients receiving R-CHOP determined the threshold of high Ki67 expression predicting 2-year progression-free survival (PFS) events using receiver operating characteristic (ROC) curve. Then we initiated a prospective, observational study evaluating DA-EPOCH-R therapy for DLBCL with high Ki67 expression, and selected a retrospective R-CHOP cohort to perform a propensity score-matched (PSM) analysis. Primary endpoints were PFS and overall survival (OS).

### Results

A Ki67 threshold of 82.5% was determined using ROC in the retrospective analysis of 376 DLBCL cases, thus the threshold was ultimately set at 80%. A total of 29 patients receiving DA-EPOCH-R were enrolled in the prospective study, the enrollment was terminated due to the conditional power and a slow accrual. Following 1:1 PSM, no PFS (HR 0.93, 95% CI: 0.47–1.84, p = 0.83) and OS (HR 1.28, 95% CI: 0.42–3.83, p = 0.66) benefit of DA-EPOCH-R versus R-CHOP was observed, with a nearly significantly higher frequency of grade ≥ 3 hematological adverse events (AEs) observed in the DA-EPOCH-R group (p = 0.052). Notably, only 37.9% (11/29) of patients executed dose escalation, which indicated a worse tolerance to this intensified regimen of the East Asians.

**Data availability statement:** The data used in this study cannot be shared publicly because it contains sensitive participant information. These restrictions were imposed by the Ethics Committee of Cancer Hospital, Chinese Academy of Medical Sciences. Interested researchers can request data from the corresponding author or the Ethics Committee of Cancer Hospital via cancergcp@163.com.

**Funding:** This study was supported by The Lymphoma Research Fund of China Anti-Cancer Association (grant id was not applicable). There was no additional external funding received for this study. The grant recipient was Peng Liu. The funders had no role in study design, data collection and analysis, decision to publish, or preparation of the manuscript.

**Competing interests:** NO authors have competing interests.

**Abbreviations:** DLBCL, diffuse large B-cell lymphoma; DA-EPOCH-R, dose-adjusted EPOCH-R; PFS, progression-free survival; AEs, adverse events; NHL, non-Hodgkin lymphoma; GCB, germinal center B-cell-like; ABC, activated B-cell; CR, complete response; CRR, complete response rate; ORR, overall response rate; OS, overall survival; DEL, double expressor lymphoma; D/THL, double/triple hit lymphoma; IHC, immunohistochemistry; SUVmax, maximum standard uptake value; ECOG PS, Eastern Cooperative Oncology Group Performance Status; WBC, white blood cell; HB, hemoglobin; ANC, absolute neutrophil count; PLT, platelet; ULN, upper limit of normal; CT, computed tomography; PET-CT, positron emission tomography - computed tomography; CTX, cyclophosphamide; SAEs, serious adverse events; IPI, international prognostic index; LDH, lactate dehydrogenase; BTKi, Bruton tyrosine kinase inhibitor; PSM, propensity score-matched; HR, hazard ratio; R/R, relapsed/refractory; CAR-T, chimeric antigen receptor T-Cell; G-CSF, granulocyte colony-stimulating factor; PEG-G-CSF, polyethylene glycolated granulocyte colony-stimulating factor; CI, confidence interval.

## Conclusions

No significant survival benefit from DA-EPOCH-R treatment was observed for DLBCL with high Ki67 expression, which may be attributed to the poorer tolerance in the East Asian population.

## 1. Introduction

As the most prevalent type of all non-Hodgkin lymphomas (NHLs) [1], diffuse large B-cell lymphoma (DLBCL) comprises three heterogeneous subtypes: germinal center B-cell-like (GCB), activated B-cell (ABC), and unclassified category based on a classification of cell origin through the gene expression. Clinically, the Hans algorithm has been commonly applied to subdivide DLBCL into two major subgroups: GCB and non-GCB (encompassing ABC subtype and most unclassified cases) [2]. The first-line R-CHOP treatment (cyclophosphamide, anthracyclines, vinblastine, prednisone, and rituximab) remains suboptimal, with about 20% of patients exhibiting primarily refractory disease and 30% relapsing [3]. Pola-R-CHP regimen exhibited a superior progression-free survival (PFS) benefit in the phase III POLARIX study, but the overall survival (OS) benefit had not reached a statistical borderline in the 5-year update analysis [4]. Patients who experience first-line treatment failure, especially refractory cases, usually have a poor prognosis [5–6]. Although the prognosis of relapsed/refractory (R/R) DLBCL improved substantially since the introduction of new therapies such as chimeric antigen receptor T-cell (CAR-T) [7–8], its application is constrained by various factors, including manufacturing challenges, limited accessibility, and economic barriers [9]. Furthermore, a proportion of patients still relapse after the CAR-T treatment [10]. Thus, it is essential to investigate intensified therapies to improve first-line PFS, particularly in those with high risk.

Given the limitations of R-CHOP, significant efforts trying to enhance the treatment efficacy in the first-line setting have been made in the past. Dose-adjusted, 96-hour infusional etoposide, cyclophosphamide, doxorubicin, vincristine, prednisone, and rituximab (DA-EPOCH-R) regimen has exhibited a promising efficacy in high-risk DLBCL patients [11], however, it did not show additional benefit except increased toxicity versus R-CHOP in CALGB 50303, a phase III trial [12]. Therefore, current exploration of this intensive regimen mainly focuses on identifying specific high-risk subgroups that may benefit. Double-expressor lymphoma (DEL), a DLBCL subtype with concurrent MYC and BCL2 overexpression, typically exhibits a poor prognosis [13] but derives no additional benefit from first-line DA-EPOCH-R compared with R-CHOP [14]. Double/triple hit lymphoma (D/THL), an advanced lymphoma type harboring rearrangements of MYC, BCL2 and/or BCL6, was previously recognized as a subtype of DLBCL but as a distinct lymphoma type nowadays. D/THL is correlated with a poor prognosis [15], and several studies demonstrated superior PFS in DHL patients receiving first-line DA-EPOCH-R therapy versus R-CHOP [16–17]. Furthermore, there is currently insufficient evidence to identify specific DLBCL patient subgroups that may benefit from DA-EPOCH-R.

As a classical marker of the cell proliferation, nuclear protein Ki67 encoded by the MKI67 gene functions in the transcription of ribosomal RNA [18]. Maximum standard uptake value (SUVmax) reflecting tumor activity is positively correlated with Ki67 in NHLs [19], a meta-analysis also confirmed a negative correlation between the high expression of Ki67 and the prognosis in DLBCL [20]. The R-EPOCH regimen achieved superior PFS and OS compared to the R-CHOP regimen in DLBCL patients with high Ki67 expression in a previous study [21], and DA-EPOCH may be more effective against rapidly dividing tumors, which is consistent with in vitro models that showed an increased sensitivity of dividing cells to DNA-damaging agents [22]. Therefore, we hypothesized that DLBCL with high Ki67 expression might benefit from DA-EPOCH-R. Given its widespread clinical availability, we initiate this study to investigate whether high Ki67 expression identifies a subgroup that may derive additional benefits from DA-EPOCH-R therapy.

## 2. Materials and methods

### 2.1 Study design

The study was designed in two phases: First, a retrospective analysis of DLBCL patients receiving R-CHOP as first-line treatment from February 1, 2012 to January 1, 2016 was carried out to determine the prognosis risk and the threshold of high Ki67 expression. Inclusion criteria of the retrospective analysis included confirmed pathological DLBCL diagnosis, availability of complete clinical data (general information, laboratory examination results, treatment, imaging examination results and follow-up). All identifying information was de-identified and anonymized prior to data analysis. Subsequently, we conducted a prospective, observational study evaluating DA-EPOCH-R for untreated DLBCL patients with high Ki67 expression from August 1, 2017 to September 30, 2022. This study was approved by the Ethics-Committee of Cancer Hospital, Chinese Academy of Medical Sciences (Number: 17–119/1375), and all subjects in the prospective study provided written informed consent. The procedures in this study were in accordance with the Helsinki Declaration [23]. Data were accessed for research purposes on 01/12/2025.

### 2.2 Ki67 immunohistochemistry (IHC) and threshold

Immunohistochemical staining for Ki67 was performed using an automated immunostainer (Roche Ventana platform) and a monoclonal antibody against Ki67 (clone GM027; GeneTech, Shanghai, China). In the retrospective cohort, the cut-off threshold for high Ki67 expression was defined by receiver operating characteristic (ROC) curve predicting the 2-year PFS. Its prognostic value was evaluated by the log-rank test, then validated using the multivariable Cox regression.

### 2.3 Inclusion and exclusion criteria of the prospective trial

Main eligible conditions included: confirmed pathological diagnosis of DLBCL (2016 World Health Organization classification of lymphoid neoplasm) [24]; Ki67 ≥ determined threshold; patients who are scheduled to receive first-line DA-EPOCH-R treatment; at least one measurable lesion; age ≥ 18 years old; no previous systemic chemotherapy; no contra-indication to rituximab (such as active hepatitis B); all disease stages; Eastern Cooperative Oncology Group Performance Status (ECOG PS) ≤ 2; hematological examination meets chemotherapy criteria: white blood cell (WBC) ≥ 4.0x$10^9$/L, absolute neutrophil count (ANC) ≥ 2x$10^9$/L, platelet (PLT) ≥ 100x$10^9$/L, hemoglobin (HB) ≥ 90 g/L (bone marrow invasion: WBC ≥ 3x$10^9$/L, ANC ≥ 1.5x$10^9$/L, PLT ≥ 75x$10^9$/L, HB ≥ 80g/L); normal liver function or tolerable abnormalities (alanine aminotransferase or aspartate aminotransferase ≤ 2.5 upper limit of normal [ULN]; total bilirubin ≤ 1.5 ULN); normal renal function (within the normal range of creatinine). Main exclusion conditions included central nervous system invasion, high-grade B-cell lymphoma, CD5-positive DLBCL, transformed lymphoma, bone marrow involvement (to ensure a tolerance to high-dose chemotherapy), and other serious concomitant diseases and patients who were considered unfit by researchers.

## 2.4 Propensity score-matched analysis

For comparison with the R-CHOP regimen, we selected a retrospective patients cohort with high Ki67 expression and R-CHOP treatment in the enrollment time range of the prospective cohort and performed a propensity score-matched (PSM) analysis to enable a more balanced comparison between the two therapies in this high-risk subgroup. PSM was conducted using the method of optimal, 1:1 matching to balance baseline characteristics between the two groups. The characteristics included in the PSM were international prognostic index (IPI), DEL, sex, and subtype (GCB or non-GCB).

## 2.5 Procedures

Since this study was observational in nature, the treatment regimens were not randomized but were chosen by patients in consultation with their physicians, who provided treatment recommendations. In our institution, treatment protocols are generally administered as follows: DA-EPOCH-R regimen: initial dose: etoposide 50 mg/m²/d, 96-hour continuous infusion on day 1–4; doxorubicin 10 mg/m²/d, 96-hour continuous infusion on day 1–4; vincristine 0.4 mg/m²/d, 96-hour continuous infusion on day 1–4; cyclophosphamide (CTX) 750 mg/m² infusion, day 5; prednisone 60 mg/m²/d oral, day 1–5; rituximab 375 mg/m² infusion, day 0; granulocyte colony-stimulating factor (G-CSF) 5 µg/kg/d subcutaneous injection on day 6 until ANC > 5x10⁹/L or polyethylene glycolated granulocyte colony-stimulating factor (PEG-G-CSF) 100 µg/kg subcutaneous injection on day 7 (every 3 weeks). Adjust the dose according to the nadir ANC checked twice a week: if the nadir ANC ≥ 0.5x10⁹/L, the doses of etoposide, anthracyclines and CTX in the next cycle would be increased by 20%; if the nadir ANC ˪0.5x10⁹/L but duration <1 week, the doses of etoposide, anthracyclines and CTX in the next cycle would remain unchanged; if the nadir ANC < 0.5x10⁹/L and duration >1 week, the doses of etoposide, anthracyclines and CTX in the next cycle would be reduced by 20%. R-CHOP regimen: CTX infusion 750 mg/m², day 1; doxorubicin infusion 50 mg/m², day 1 or epirubicin infusion 40 mg/m²/d, day 1–2; vincristine infusion 1.4 mg/m² (up to 2 mg), day 1; prednisone oral 100 mg, day 1–5; rituximab infusion 375 mg/m², day 0 (every 3 weeks).

Pre-treatment evaluations comprised standard laboratory examination and positron emission tomography-computed tomography (PET-CT), while the whole-body computed tomography (CT) was permitted given the early initiation of the study. Immunohistochemical biopsy samples were mandatory before treatment. Treatment would be terminated in cases of tumor progression or unacceptable serious adverse events (SAEs). A consolidative radiotherapy was permitted after the multidisciplinary team discussion. Follow-up was conducted at 3-month intervals during the first year, 3–6 months between years 1–3, and 6–12 months after the third year. Tumor response was assessed by whole-body CT or PET-CT every 2 treatment cycles, with efficacy evaluation based on the 2014 Lugano criteria [25].

## 2.6 Outcomes

The primary endpoints of this study included PFS (the time from treatment initiation to either progression, relapse, or death) and OS (the time from treatment initiation to all-cause death). Secondary endpoints included tumor response and safety.

## 2.7 Statistical analysis

The sample size calculation of the prospective study utilized the exponential distribution assumption for survival times. Given the R-EPOCH regimen achieved a 2-year PFS rate more than 85% in DLBCL patients with high Ki67 expression [21], we anticipate that DA-EPOCH-R could further improve this outcome based on the biological rationale, with a hazard ratio (HR) of 0.3 compared to historical controls, corresponding to an expected improvement in 2-year PFS rate from 70% of R-CHOP to 90%. Using a one-sided significance level of α = 0.05 and a power (1-β) of 80%, the theoretical sample size was determined to be 60 patients. Considering an estimated 10% dropout rate, the sample size of the prospective study was set at 67 patients. However, at the interim analysis, enrollment was terminated due to the conditional power failing

to reach 10% and a slow accrual. Patients' characteristics and adverse events (AEs) were summarized by descriptive statistics. The Clopper-Pearson method was applied to estimate the 95% confidence interval (CI) for the response rate. The difference between the survival curves was verified using the log-rank test. AEs were documented and reported in accordance with the revised NCI General Terminology Standard for Adverse Events Version 5.0. Statistical analysis was conducted by software SPSS (version 26.0) and R (version 4.5.1). A P-value < 0.05 was considered statistically significant.

## 3. Results

### 3.1 Baseline characteristics

In the retrospective analysis part, 376 DLBCL patients receiving first-line R-CHOP from February 2012 to January 2016 were included (baseline characteristics of the patients were demonstrated in S1 Table), an optimal Ki67 cut-off predicting 2-year PFS events determined by ROC was 82.5% (S1 Fig, AUC = 0.65; 95% CI: 0.55–0.74). Taking clinical practice into consideration, the optimal threshold was ultimately set at 80%. Patients with Ki67 ≥ 80% exhibited a significantly shorter PFS compared with those with Ki67 < 80% (log-rank p = 0.0037, Fig 1A), and validated by the multivariable Cox regression (HR 1.67, 95% CI: 1.05–2.65, p = 0.029, Fig 1B), supporting the classification of Ki67 ≥ 80% as a high-risk DLBCL subgroup. Next, from August 2017 to September 2022, a total of 29 patients with Ki67 ≥ 80% and DA-EPOCH-R treatment were enrolled in the prospective study. In the entire cohort, the GCB subtype accounted for 69.0%, while 37.9% of patients had DEL and 65.5% had advanced-stage disease. A retrospective cohort including 165 patients with high Ki67 expression who received R-CHOP during the enrollment period of the prospective cohort were recorded. After 1:1 PSM, all baseline characteristics between the matched groups were well-balanced (Table 1). Median treatment cycles of the two treatment groups (after PSM) were both 6 (range: 4–8).

### 3.2 Dose level and response

The median dose level across all treatment cycles was level 1 (100%) (range: 51–249%). Notably, only 37.9% (11/29) of patients executed dose escalation according to the protocol among all subjects receiving DA-EPOCH-R (Fig 2A), significantly lower than the proportion reported in the previous study [26]. 37.9% (11/29) maintained the initial dose, and 24.1% (7/29) required dose reductions. Dose escalation was uniformly precluded by hematologic toxicity. Elderly patients (age ≥ 60 years old) did not exhibit a lower dose-escalation rate compared with non-elderly patients (22.2% vs. 45.0%; Fisher's exact test p = 0.412).

DA-EPOCH-R treatment group (n = 29) achieved a complete response rate (CRR) of 51.7% (15/29; 95% CI: 32.4–71.1%) and an ORR of 93.1% (27/29; 95% CI: 83.3–100) (Fig 2B), the CRR of the dose-escalation group was not statistically higher (54.5% vs. 44.4%, p = 0.597). Two patients with no response maintained the initial dose. In contrast, the control R-CHOP group achieved a CRR of 55.2% (16/29; 95% CI: 35.9–74.4%) and an ORR of 93.1% (27/29; 95% CI: 83.3–100). CRR in the DA-EPOCH-R treatment group was not superior to that in the R-CHOP group (p = 0.792).

### 3.3 Survival

As of December 1, 2025, after a median follow-up time of 46.1 months (95% CI: 35.0–57.3), 16 patients experienced disease progression or relapse, with 6 cases occurring in the first year after the treatment initiation. The median PFS of the DA-EPOCH-R treatment group was 37.4 months (95% CI: 30.2–44.7), and the PFS rates at the 1 and 2 years were 79.3% (95% CI: 65.9–95.5) and 65.5% (95% CI: 50.3–85.3), respectively. After PSM analysis, no statistical difference in PFS between the two treatment groups was demonstrated (HR 0.93, 95% CI: 0.47–1.84, p = 0.83, Fig 3). Subgroup analysis suggested trends toward prolonged PFS with DA-EPOCH-R treatment in the bulky disease (HR 0.39, 95% CI: 0.10 − 1.50, p = 0.168) and stage I-II (HR 0.47, 95% CI: 0.12 − 1.90, p = 0.291) subgroups, while patients with Ki67 ≥ 90% did not exhibit a PFS benefit from DA-EPOCH-R (HR 0.94, 95% CI: 0.30 − 2.94, p = 0.908, S2 Fig).

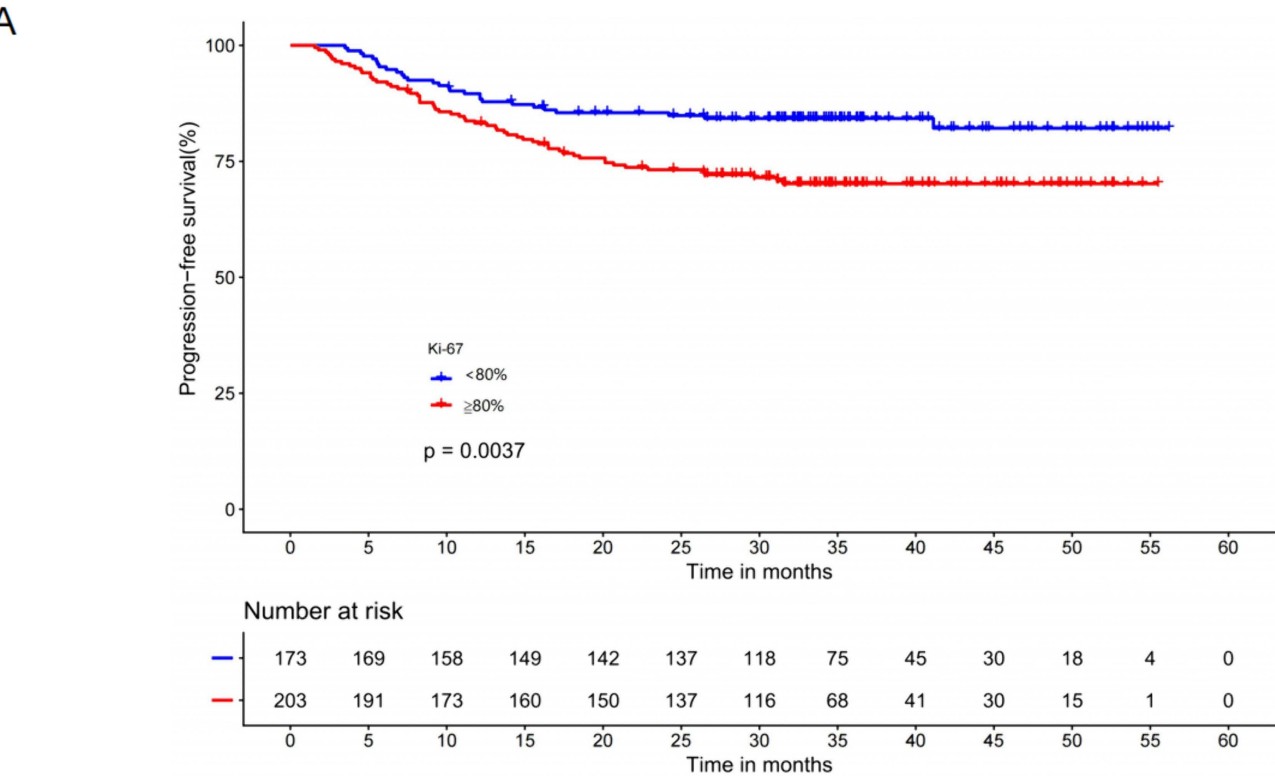

**Fig 1. Progression-free survival analysis of the retrospective cohort receiving R-CHOP. A.** Kaplan-Meier curves of PFS stratified by Ki67 and log-rank test. **B.** Multivariable Cox regression for PFS of the retrospective cohort.

**Table 1. Baseline characteristics of DLBCL patients with Ki67 ≥ 80%.**

| | Before PSM (n[%]) | | | After PSM (n[%]) | | |
|---|---|---|---|---|---|---|
| | DA-EPOCH-R (n = 29) | R-CHOP (n = 165) | p | DA-EPOCH-R (n = 29) | R-CHOP (n = 29) | p |
| Male | 16 (55.2) | 89 (53.9) | 0.902 | 16 (55.2) | 15 (51.7) | 0.792 |
| ECOG PS ≥ 1 | 15 (51.7) | 89 (53.9) | 0.825 | 15 (51.7) | 15 (51.7) | 1.000 |
| GCB | 20 (69.0) | 79 (47.9) | 0.036 | 20 (69.0) | 22 (75.9) | 0.557 |
| Extranodal involvement > 1 | 13 (44.8) | 96 (58.2) | 0.181 | 13 (44.8) | 14 (48.3) | 0.792 |
| Advanced stage | 19 (65.5) | 55 (33.3) | 0.001 | 19 (65.5) | 18 (62.1) | 0.785 |
| Age > 60 | 9 (31.0) | 65 (39.4) | 0.393 | 9 (31.0) | 10 (34.5) | 0.780 |
| Bulky disease (>7.5 cm) | 8 (27.6) | 23 (13.9) | 0.064 | 8 (27.6) | 6 (20.7) | 0.539 |
| LDH > normal range | 19 (65.5) | 63 (38.2) | 0.006 | 19 (65.5) | 20 (69.0) | 0.780 |
| DEL | 11 (37.9) | 68 (41.2) | 0.740 | 11 (37.9) | 10 (34.5) | 0.785 |
| IPI ≥ 3 | 16 (55.2) | 62 (37.6) | 0.075 | 16 (55.2) | 16 (55.2) | 1.000 |
| BCL-2/BCL-6 rearrangement | 3 (10.3) | 14 (8.5) | 0.724 | 3 (10.3) | 5 (17.2) | 0.706 |

*LDH: lactate dehydrogenase; ECOG PS: Eastern Cooperative Oncology Group Performance Status; Bulky disease: primary lesion >7.5 cm; GCB: germinal center B-cell-like; DEL: double expressor lymphoma; IPI: international prognostic index.

After disease progression or relapse, 2 patients received CAR-T, and 5 patients underwent stem-cell transplantations. In the DA-EPOCH-R treatment group, the median OS was not reached, with a HR of 1.28 (95% CI: 0.42–3.83) compared with the R-CHOP group (p = 0.66, Fig 4).

### 3.4 Adverse events

In the DA-EPOCH-R treatment group, 100% of patients experienced grade ≥ 3 AEs (Table 2). Grade ≥ 3 hematological and non-hematological AEs were reported in 100.0% and 51.7% of cases, respectively. The frequency of grade ≥ 3 hematological AEs was nearly significantly higher in the DA-EPOCH-R group than that in the R-CHOP group (Fisher's exact test p = 0.052). In contrast, no statistical difference in the frequency of grade ≥ 3 non-hematological AEs between the two treatment groups was observed (p = 0.599). No fatal toxicity was observed.

## 4. Discussion

High Ki67 proliferation index was consistently correlated with an inferior prognosis in DLBCL patients [27–29], raising a critical question of whether intensified regimens could generate additional benefits for this aggressive subgroup. Disappointingly, the enrollment of this study was terminated at the interim analysis due to the unsatisfactory conditional power, poor tolerability and slow accrual, we did not observe a survival benefit from DA-EPOCH-R for DLBCL patients with high Ki67 expression, although this finding warrants a cautious interpretation given the limited sample size of the cohort. Notably, DA-EPOCH-R regimen demonstrated poorer safety profiles in the East Asian populations, who tend to exhibit relatively lower tolerance to chemotherapy [30–32]. The worse tolerance might partially compromise its efficacy, underscoring the importance of careful patient selection based on treatment tolerance.

The design of the DA-EPOCH regimen was based on pharmacokinetic analyses, which indicated significant inter-patient variations in plasma concentrations of etoposide and doxorubicin, suggesting the need for individual dose adjustment [33]. In the initial phase II trial, DA-EPOCH treatment demonstrated a remarkable efficacy, with a CRR of 92% and an ORR of 100% [22]. However, subsequent studies failed to provide sufficient evidence supporting DA-EPOCH-R's superiority over R-CHOP regimen, leading to ongoing debate regarding the clinical value of this intensive therapeutic approach. Despite extensive investigation, no intensified regimen other than DA-EPOCH-R has accumulated sufficient evidence to challenge R-CHOP's established role as first-line therapy for DLBCL. Although PFS

A

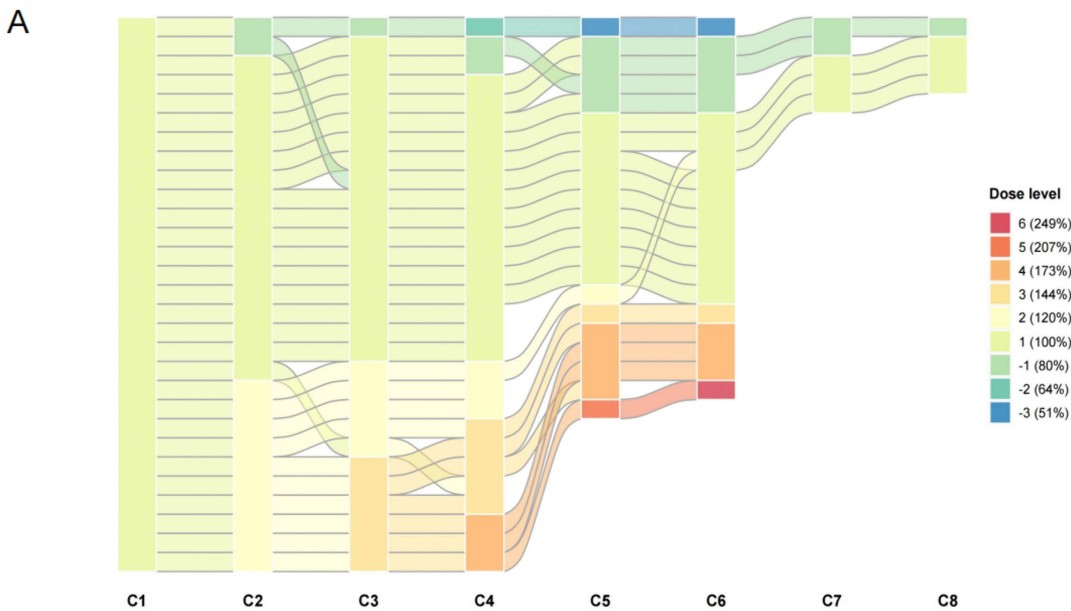

B

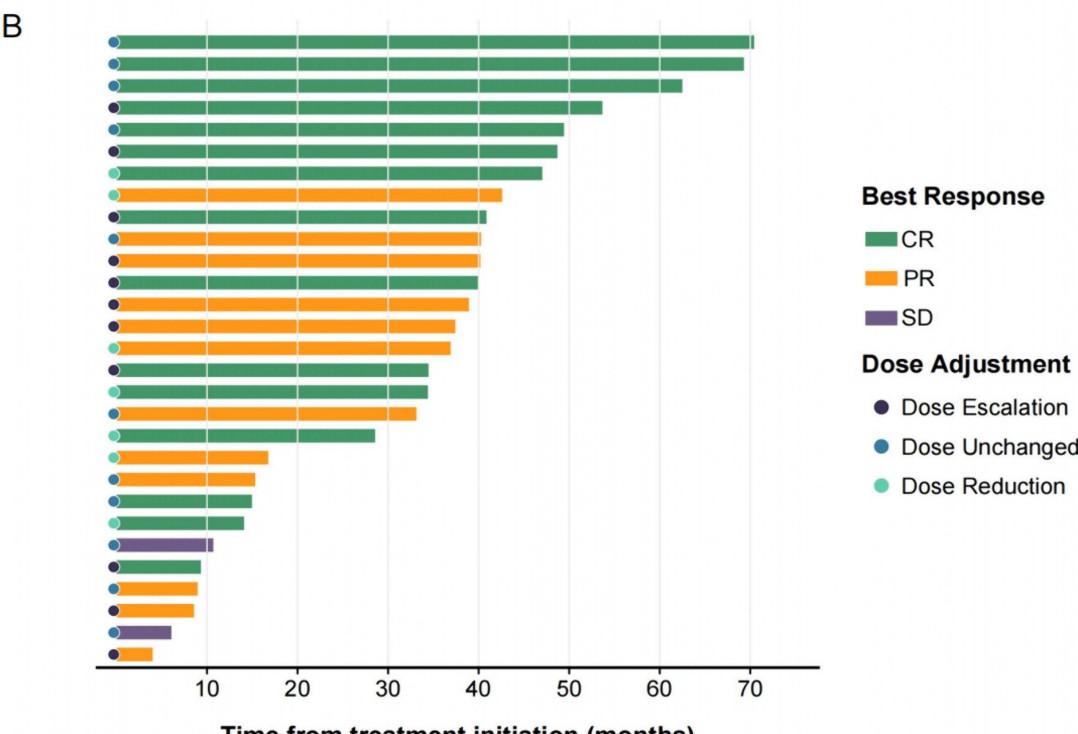

**Fig 2. Dose level and response of the prospective cohort receiving DA-EPOCH-R treatment. A.** Sankey diagram of the distribution of dose levels across treatment cycles. **B.** Swimplot of the prospective cohort. CR: complete response; PR: partial response; SD: stable disease.

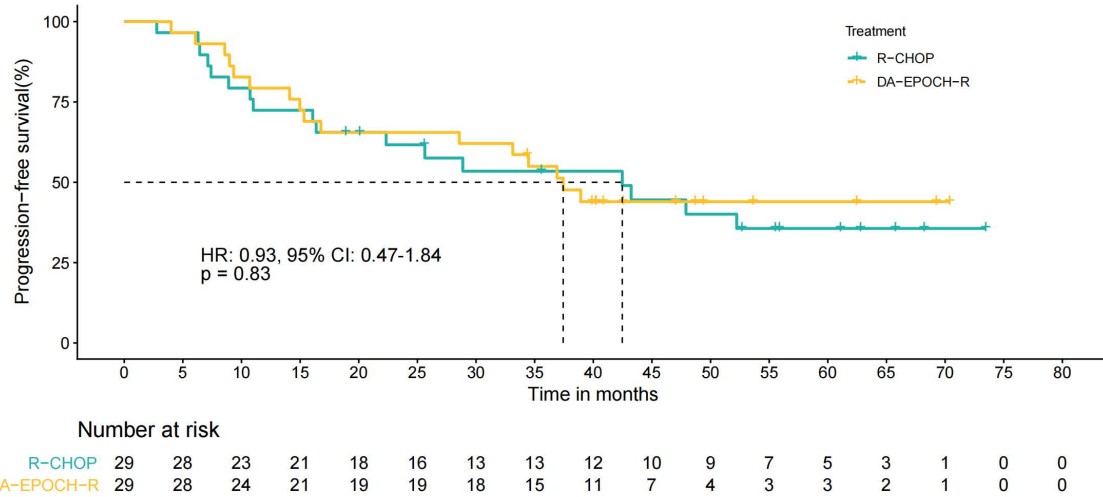

**Fig 3. PFS curves of two treatment groups.**

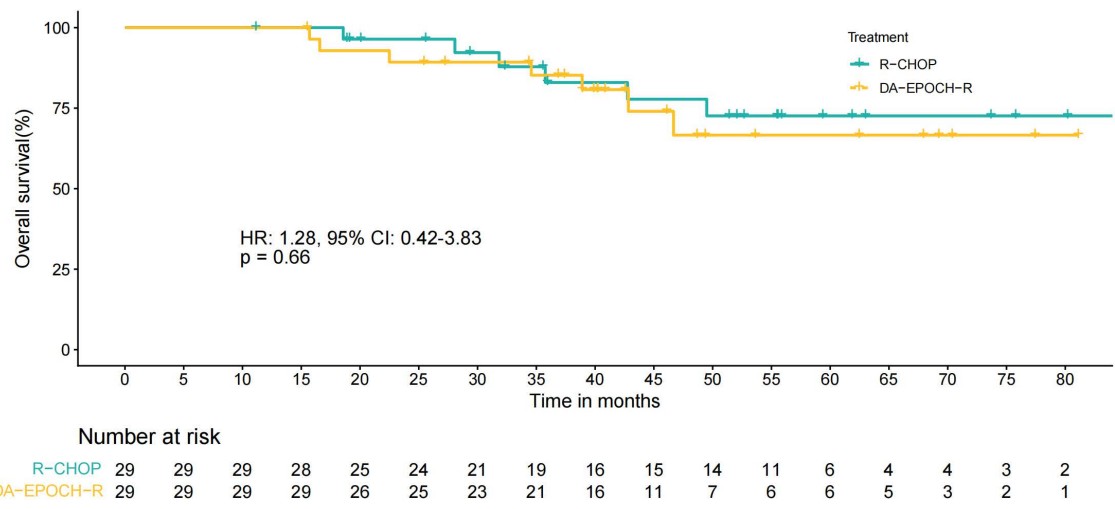

**Fig 4. OS curves of two treatment groups.**

and OS of R-ACVBP reinforcement in the phase III LNH03-2B trial were longer than those of R-CHOP treatment in low-medium risk and young DLBCL patients [34], the subsequent phase III GAINED study did not confirm the benefit [35]. Similarly, switching to higher-intensity Burkitt regimen for mid-term PET-CT positive patients after 2 cycles of R-CHOP provided no survival benefit [36]. Collectively, these findings suggest that intensified chemotherapy regimens did not represent a superior therapeutic strategy.

One alternative strategy to enhance first-line treatment efficacy for DLBCL is the incorporation of an additional targeted agent X. Pola-R-CHP treatment represents a superior and standard frontline treatment for DLBCL due to the superior PFS over R-CHOP according to the phase III POLARIX study, although the OS benefit has not been confirmed yet [4]. Due to the initiation of our study prior to the evolution of the current treatment landscape, a direct comparison between DA-EPOCH-R and Pola-R-CHP could not be conducted. Although R-CHOP plus Bruton tyrosine kinase inhibitor (BTKi)

**Table 2. Frequent (>5%) grade ≥3 AEs of each treatment group.**

| | R-CHOP | | DA-EPOCH-R | |
|---|---|---|---|---|
| | n | % | n | % |
| All grade≥3AEs | 25 | 86.2 | 29 | 100.0 |
| Hematological AEs | 24 | 82.8 | 29 | 100.0 |
| Neutropenia | 24 | 82.8 | 29 | 100.0 |
| Anaemia | 2 | 6.9 | 8 | 27.6 |
| Thrombocytopenia | 2 | 6.9 | 8 | 27.6 |
| Febrile neutropenia | 8 | 27.6 | 16 | 55.2 |
| Non-hematological AEs | 13 | 44.8 | 15 | 51.7 |
| Vomiting | 2 | 6.9 | 3 | 10.3 |
| Mucositis | 2 | 6.9 | 2 | 10.3 |
| Peripheral neuropathy | 0 | 0.0 | 6 | 20.7 |
| Increased transaminase | 2 | 6.9 | 2 | 6.9 |
| Infection | 2 | 6.9 | 3 | 10.3 |

ibrutinib could not exhibit a survival benefit in non-GCB patients in the phase III PHOENIX study, a subgroup analysis for patients ≤ 60 years old showed improvements in survival, including PFS and OS [37]. Additionally, combining lenalidomide or bortezomib with R-CHOP did not demonstrate a prognosis improvement [38–39], while adding venetoclax to R-CHOP prolonged both PFS and OS but with more frequent toxicities [40]. Importantly, molecular classification of DLBCL based on gene expression signatures enables more precise prognostic prediction and treatment selection. DLBCL could be divided into several genetic subtypes by next generation sequencing, exhibiting distinct biological characteristics [41]. The phase II GUIDANCE-01 trial classified patients into six genetic subtypes and administered targeted agent X based on subtype-specific alterations after one cycle of standard R-CHOP. This therapeutic strategy correlated with an improvement in the efficacy and survival outcomes compared to R-CHOP [42], highlighting the potential as a promising direction for future precision medicine.

Several limitations of our research must be acknowledged. First, the observational study had a limited sample size, which may introduce potential bias. Second, the low proportion of patients who successfully underwent dose escalation in our cohort might lead to a potential underestimation of the efficacy of the DA-EPOCH-R.

## 5. Conclusion

DA-EPOCH-R demonstrated neither survival benefit nor acceptable tolerability in East Asian DLBCL patients with high Ki67 expression.

## Supporting information

**S1 Fig. ROC curve analysis of Ki67 for predicting 2-year progression-free survival events in the retrospective cohort.**
(TIF)

**S2 Fig. Forest plot of subgroup analysis for PFS.**
(TIF)

**S1 Table. Baseline characteristics of the retrospective cohort.**
(DOCX)

## Acknowledgments

The authors gratefully acknowledge the invaluable cooperation of all study participants.

## Author contributions

**Conceptualization:** Peng Liu.

**Data curation:** Wenxin Jiang.

**Formal analysis:** Wenxin Jiang.

**Funding acquisition:** Peng Liu.

**Investigation:** Wenxin Jiang.

**Methodology:** Wenxin Jiang.

**Project administration:** Peng Liu.

**Resources:** Peng Liu.

**Software:** Wenxin Jiang.

**Supervision:** Peng Liu.

**Validation:** Peng Liu.

**Visualization:** Wenxin Jiang.

**Writing – original draft:** Wenxin Jiang.

**Writing – review & editing:** Peng Liu.

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
