## [Decision Letter · Decision Letter 0]

8 Apr 2026

PONE-D-26-12059Comparison of dose-adjusted EPOCH-R and R-CHOP in diffuse large B-cell lymphoma with high Ki67 expression: results from a prospective observational studyPLOS One

Dear Dr. Liu,

Thank you for submitting your manuscript to PLOS ONE. After careful consideration, we feel that it has merit but does not fully meet PLOS ONE’s publication criteria as it currently stands. Therefore, we invite you to submit a revised version of the manuscript that addresses the points raised during the review process.Please re-submit only if you can fully answer criticism from both Reviewers, and in particular those from Reviewer #1.

We look forward to receiving your revised manuscript.

Kind regards,

Francesco Bertolini, MD, PhD

Academic Editor

PLOS One

Journal Requirements:

" This study was supported by The Lymphoma Research Fund of China Anti-Cancer Association. The grant recipient was Peng Liu. The funders had no role in study design, data collection and analysis, decision to publish, or preparation of the manuscript."

3. In the online submission form, you indicated that [The data underlying this article cannot be shared publicly due to the privacy concerns and restrictions imposed by the ethics committee. The data are available from the corresponding author upon reasonable request, subject to approval and appropriate data transfer agreements, for researchers who meet the criteria for access to confidential medical data.].

Reviewers' comments:

Reviewer's Responses to Questions

**Comments to the Author**

1. Is the manuscript technically sound, and do the data support the conclusions?

Reviewer #1: No

Reviewer #2: Yes

2. Has the statistical analysis been performed appropriately and rigorously? 

Reviewer #1: No

Reviewer #2: Yes

3. Have the authors made all data underlying the findings in their manuscript fully available?

Reviewer #1: Yes

Reviewer #2: No

4. Is the manuscript presented in an intelligible fashion and written in standard English?

Reviewer #1: Yes

Reviewer #2: Yes

5. Review Comments to the Author

Reviewer #1: Jiang and colleagues reported the results of both retrospective and prospective studies comparing the therapeutic efficacy of DA-EPOCH-R and R-CHOP in patients with high Ki-67 DLBCL. This manuscript contains some interesting information; however, there are some severe problems in this manuscript.

Major comments:

1) The rationale for selecting DA-EPOCH-R for patients with Ki-67-high DLBCL is not entirely clear. Although a high Ki-67 index may reflect a highly aggressive proliferative phenotype, it remains unclear whether this alone justifies the use of DA-EPOCH-R. The authors should provide a clearer explanation of the biological and/or clinical rationale for this treatment strategy and present evidence to support it.

2) The manuscript lacks essential information regarding the study design. Specifically, the phase of the clinical trial is not specified, and key statistical parameters such as the alpha level and statistical power are not described. Without these details, it is difficult to interpret the validity of the study as a prospective clinical trial. The authors should clearly define the trial phase and provide a detailed statistical design, including assumptions for sample size calculation.

3) The prior retrospective analysis reported a 2-year PFS of approximately 75% in the patients with high KI-67 DLBCL received DA-EPOCH-R. In this context, the assumption of a 2-year PFS of 90% with DA-EPOCH-R appears overly optimistic and lacks sufficient justification. This substantial discrepancy is not addressed in the manuscript. The authors should clarify the rationale for this assumption or provide supporting evidence.

4) The proportion of GCB-type DLBCL in this cohort is relatively high (69%). Although the authors state that HGBCL cases were excluded, it is unclear whether MYC and BCL2 rearrangements were systematically assessed by FISH. Without such evaluation, the possibility that HGBCL cases were inadvertently included cannot be excluded.

Reviewer #2: This is prospective cohort study using a retrospective propensity score matched population to compare DA-EPOCH-R with R-CHOP in pts with KI67> 80%. No differences are seen in PFS / OS. The study design is solid, the manuscript well written and results are resented in a clear way. I only have a few minor suggestions for improvement:

Minor comments:

line 126: "treatment regimens were chosen by patients". Surely the patients did not choose the treatment regiments. Please adjust the wording for clarity.

line 194: Instead of "elderly" please state the exact age cut off you used, because for me it was not clear.

line 278: "high-risk" can be misinterpreted, as risk is usually defined per IPI. I Would suggest to write instead "with high Ki67 expression"

6. PLOS authors have the option to publish the peer review history of their article (what does this mean?). If published, this will include your full peer review and any attached files.

Reviewer #1: No

Reviewer #2: No

---

## [Author Response · Author response to Decision Letter 1]

17 Apr 2026

Dear editors and reviewers:

We greatly appreciate your efforts during the review process and the opportunity to make corrections and supplements, which have significantly improved the quality of our manuscript. We have revised the content according to the reviewers’ comments, with the changes clearly marked (in red font, excluding changes in reference numbers due to added citations), and have provided point-by-point explanations for the revisions.

We would like to express our sincere gratitude for your valuable time and constructive feedback. Should there be any further concerns or additional modifications required, we are more than willing to make any necessary revisions promptly.

Point-by-point response

Reviewer #1: Jiang and colleagues reported the results of both retrospective and prospective studies comparing the therapeutic efficacy of DA-EPOCH-R and R-CHOP in patients with high Ki-67 DLBCL. This manuscript contains some interesting information; however, there are some severe problems in this manuscript.

Major comments:

1)The rationale for selecting DA-EPOCH-R for patients with Ki-67-high DLBCL is not entirely clear. Although a high Ki-67 index may reflect a highly aggressive proliferative phenotype, it remains unclear whether this alone justifies the use of DA-EPOCH-R. The authors should provide a clearer explanation of the biological and/or clinical rationale for this treatment strategy and present evidence to support it.

Response: We fully agree with your reasonable concern. We agree that a high Ki-67 index alone, as a measure of proliferation, is insufficient to justify the selection of a specific chemotherapy regimen. In the revised manuscript, we have provided a clearer biological and clinical rationale for choosing DA-EPOCH-R in this setting (line 82-87), which we summarize as follows:

1.Biological Rationale: A previous study indicated that unlike traditional chemotherapy (e.g., CHOP), DA-EPOCH utilizes a 96-hour continuous infusion, which may be more effective against rapidly dividing tumors and are consistent with in vitro models that showed an increased sensitivity of dividing cells to DNA-damaging agents [Blood. 2002 Apr 15;99(8):2685-93. doi: 10.1182/blood.v99.8.2685.]. Therefore, we hypothesized that DLBCL with high Ki67 expression might benefit from DA-EPOCH-R.

2.Clinical Evidence: The R-EPOCH regimen achieved superior PFS and OS compared to the R-CHOP regimen in DLBCL patients with high Ki67 expression in a previous study [Oncotarget. 2016 May 10;7(27):41242-41250. doi: 10.18632/oncotarget.9271], we hypothesized that the DA-EPOCH-R regimen might offer greater benefits for this patient population.

We have added the following references and corresponding text to the manuscript to support this rationale.

2) The manuscript lacks essential information regarding the study design. Specifically, the phase of the clinical trial is not specified, and key statistical parameters such as the alpha level and statistical power are not described. Without these details, it is difficult to interpret the validity of the study as a prospective clinical trial. The authors should clearly define the trial phase and provide a detailed statistical design, including assumptions for sample size calculation.

Response: We fully agree with your valuable comment. The design of this trial followed standard sample size and statistical analysis procedures. However, we inappropriately omitted certain methodological details during writing. We have now re-added the alpha level and statistical power used in the study design to the Statistical Analysis section (line 164-165, “Using a one-sided significance level of α=0.05 and a power (1-β) of 80%”). Since this is an observational study, the concept of clinical trial phases does not apply.

3) The prior retrospective analysis reported a 2-year PFS of approximately 75% in the patients with high KI-67 DLBCL received DA-EPOCH-R. In this context, the assumption of a 2-year PFS of 90% with DA-EPOCH-R appears overly optimistic and lacks sufficient justification. This substantial discrepancy is not addressed in the manuscript. The authors should clarify the rationale for this assumption or provide supporting evidence.

Response: We fully agree with and understand your considerate comments. We also took into account the current state of research at the time when designing the study, but we did not clearly mention this in the manuscript. In a previous study, the R-EPOCH regimen (not DA-EPOCH-R) achieved a 2-year PFS rate more than 85% in DLBCL patients with high Ki67 expression [Oncotarget. 2016 May 10;7(27):41242-41250. doi: 10.18632/oncotarget.9271], thus we anticipate that DA-EPOCH-R could further improve this outcome based on the biological rationale. The relevant statements have now been re-added to the manuscript (line 161-164, “Given the R-EPOCH regimen achieved a 2-year PFS rate more than 85% in DLBCL patients with high Ki67 expression , we anticipate that DA-EPOCH-R could further improve this outcome based on the biological rationale”).

4) The proportion of GCB-type DLBCL in this cohort is relatively high (69%). Although the authors state that HGBCL cases were excluded, it is unclear whether MYC and BCL2 rearrangements were systematically assessed by FISH. Without such evaluation, the possibility that HGBCL cases were inadvertently included cannot be excluded.

Response: We fully understand this reasonable concern. All patients in the cohort underwent FISH testing to rule out HGBCL. However, patients with a single MYC or BCL2 rearrangement were not excluded, as they accounted for a relatively small proportion of the cohort. We have added this information back into the baseline characteristics (Table 1).

Reviewer #2: This is prospective cohort study using a retrospective propensity score matched population to compare DA-EPOCH-R with R-CHOP in pts with KI67> 80%. No differences are seen in PFS / OS. The study design is solid, the manuscript well written and results are resented in a clear way. I only have a few minor suggestions for improvement:

Minor comments:

line 126: "treatment regimens were chosen by patients". Surely the patients did not choose the treatment regiments. Please adjust the wording for clarity.

Response: We apologize for any ambiguity caused by unclear expressions in the manuscript. We agree that the original wording “chosen by patients based on physician recommendations” was imprecise and could be misinterpreted as patients independently selecting specific treatment regimens. In fact, as a prospective observational study, the decision-making process was as follows: physicians first provided recommended treatment options based on clinical guidelines and their judgment; patients then made the final choice after being fully informed, in consultation with their physicians, the choice was strongly guided by physician recommendations. We have revises the sentence as “the treatment regimens were not randomized but were chosen by patients in consultation with their physicians, who provided treatment recommendations.” (line 131-132)

line 194: Instead of "elderly" please state the exact age cut off you used, because for me it was not clear.

Response: Thank you for your correction. This was indeed an inaccuracy in our writing. We have now added an explanation of the age cutoff value (age≥60 years old) to the manuscript (line 201).

line 278: "high-risk" can be misinterpreted, as risk is usually defined per IPI. I Would suggest to write instead "with high Ki67 expression"

Response: Thank you for your correction. We have revised the ambiguous term 'high risk,' which will undoubtedly enhance the scientific rigor of the manuscript (line 284-285).

---

## [Decision Letter · Decision Letter 1]

7 May 2026

Comparison of dose-adjusted EPOCH-R and R-CHOP in diffuse large B-cell lymphoma with high Ki67 expression: results from a prospective observational study

PONE-D-26-12059R1

Dear Dr. Liu,

We’re pleased to inform you that your manuscript has been judged scientifically suitable for publication and will be formally accepted for publication once it meets all outstanding technical requirements.

Kind regards,

Francesco Bertolini, MD, PhD

Academic Editor

PLOS One

Additional Editor Comments (optional):

Reviewers' comments:

Reviewer's Responses to Questions

**Comments to the Author**

1. If the authors have adequately addressed your comments raised in a previous round of review and you feel that this manuscript is now acceptable for publication, you may indicate that here to bypass the “Comments to the Author” section, enter your conflict of interest statement in the “Confidential to Editor” section, and submit your "Accept" recommendation.

Reviewer #1: All comments have been addressed

Reviewer #2: All comments have been addressed

2. Is the manuscript technically sound, and do the data support the conclusions?

Reviewer #1: Yes

Reviewer #2: Yes

3. Has the statistical analysis been performed appropriately and rigorously? 

Reviewer #1: Yes

Reviewer #2: Yes

4. Have the authors made all data underlying the findings in their manuscript fully available?

Reviewer #1: Yes

Reviewer #2: No

5. Is the manuscript presented in an intelligible fashion and written in standard English?

Reviewer #1: Yes

Reviewer #2: Yes

6. Review Comments to the Author

Reviewer #1: The manuscript has been revised appropriately in response to the reviewers’ comments, and the overall quality of the manuscript has been substantially improved.　Therefore, in its current form, this manuscript is considered acceptable for publication.

Reviewer #2: No more comments, I appriociate you for taking my suggestions into consoderation. The manuscript remains of limited clinical significance, however is suitable for publication.

7. PLOS authors have the option to publish the peer review history of their article (what does this mean?). If published, this will include your full peer review and any attached files.

Reviewer #1: No

Reviewer #2: No

---

## [Editor Report · Acceptance letter]

PONE-D-26-12059R1

PLOS One

Dear Dr. Liu,

I'm pleased to inform you that your manuscript has been deemed suitable for publication in PLOS One. Congratulations! Your manuscript is now being handed over to our production team.

Kind regards,

on behalf of

Dr. Francesco Bertolini

Academic Editor

PLOS One